# CRISPR GUARD protects off-target sites from Cas9 nuclease activity using short guide RNAs

Matthew A. Coelho [1,3✉], Etienne De Braekeleer[1], Mike Firth[1], Michal Bista[1], Sebastian Lukasiak[1], Maria Emanuela Cuomo [2] & Benjamin J. M. Taylor [1✉]

Precise genome editing using CRISPR-Cas9 is a promising therapeutic avenue for genetic diseases, although off-target editing remains a significant safety concern. Guide RNAs shorter than 16 nucleotides in length effectively recruit Cas9 to complementary sites in the genome but do not permit Cas9 nuclease activity. Here we describe CRISPR Guide RNA Assisted Reduction of Damage (CRISPR GUARD) as a method for protecting off-targets sites by co-delivery of short guide RNAs directed against off-target loci by competition with the on-target guide RNA. CRISPR GUARD reduces off-target mutagenesis while retaining on-target editing efficiencies with Cas9 and base editor. However, we discover that short guide RNAs can also support base editing if they contain cytosines within the deaminase activity window. We explore design rules and the universality of this method through in vitro studies and high-throughput screening, revealing CRISPR GUARD as a rapidly implementable strategy to improve the specificity of genome editing for most genomic loci. Finally, we create an online tool for CRISPR GUARD design.

[1] Discovery Sciences, R&D, AstraZeneca, Cambridge, UK. [2] Oncology R&D, AstraZeneca, Cambridge, UK. [3] Present address: Wellcome Sanger Institute, Hinxton, Cambridge CB10 1RQ, UK. ✉email: matthew.coelho@sanger.ac.uk; benjamin.taylor@astrazeneca.com

A major limitation of gene editing with CRISPR-Cas9 systems is off-target mutagenesis through guide RNA (gRNA) directed binding at closely matched sequences in the genome[1–10]. Although gRNA design algorithms are under continual refinement[11,12], off-target activity can be unavoidable when the gRNA window is restricted to a narrow genomic location, such as for therapeutic correction of disease causing mutations[5,13]. Protein engineering strategies have resulted in higher-fidelity Cas9 variants that reduce, but do not eliminate, off-target mutations; such variants can also show impaired on-target activity[14]. While off-target mutations remain a persistent problem, they can be readily detected using a variety of methodologies[2,7,15–17]. A system that is not restricted to a Cas9 variant and further reduces or eliminates particularly detrimental off-target mutations whilst retaining on-target activity would be invaluable for therapeutic strategies. Here, we develop CRISPR GUARD, a methodology that aims to block mismatched gRNAs from binding off-target sites through competition with an inactive Cas9 complex. The inactive complexes are generated by Cas9 binding short gRNAs, or GUARD RNAs, with perfect complementary to the off-target site; gRNAs shorter than 16 nucleotides (nt) in length can direct Cas9 binding but do not support nuclease activity[18–20]. This method can be adapted to use catalytically inactive Cas9 (dCas9), related RNA-guided nucleases, base editors or other sequence-specific DNA binding proteins to form inert complexes occupying off-target loci to render them inaccessible (Fig. 1).

## Results

### In vitro protection of Cas9 off-targets with CRISPR GUARD.
We tested the concept of CRISPR GUARD by measuring the binding kinetics of a perfectly complementary 15-nt GUARD RNA versus a mismatched gRNA to an immobilised DNA off-target template for a gRNA targeting *VEGFA* with four mismatches[21–23]. Bio-layer interferometry (BLI) revealed comparable off-target association kinetics for catalytically inactive Cas9 complexed with GUARD RNA or mismatched gRNA, with 29 ± 4% slower binding for the GUARD RNA (Fig. 2a), suggesting that competition at off-target loci is feasible. To test the possibility of DNA protection by CRISPR GUARD, we selected an array of potential GUARD RNA designs for in vitro Cas9 DNA cleavage assays. We considered GUARD RNA lengths of 14-nt or 15-nt, or a full length 20-nt design with only 15-nt of target complementarity (15-nt+ spacer). GUARD RNAs were designed as competitive molecules that are truncated versions of the on-target gRNA but incorporate mismatches found at the off-target site (Fig. 2b). Alternatively, we designed proximal GUARD RNAs that bind flanking regions of the off-target site, speculating that

the reduced sequence homology would reduce competition at the on-target site, but still block the protospacer and protospacer adjacent motif (PAM). We termed these designs as competitive and non-competitive GUARD RNAs, respectively (Fig. 2b). Firstly, we confirmed that GUARD RNAs cannot direct nuclease activity by in vitro assessment of on-target and off-target cleavage using short purified DNA targets. None of the GUARD RNAs directed DNA cleavage when complexed with Cas9 alone (Fig. 2c). Cas9 complexed with *VEGFA* gRNA showed robust cleavage of both on-target, and off-target *CAVIN4* DNA. However, addition of non-competitive GUARD RNAs failed to protect the off-target site. Significant protection was only observed using a 15-nt competitive GUARD RNA, but on-target cleavage was also impaired suggesting competition at both sites (Fig. 2c). Competition with the on-target gRNA may be particularly pertinent when there are very few mismatches, or when the mismatches of the gRNA cannot be incorporated in the truncated GUARD RNA design.

Cas9-gRNA complexes show rapid target binding and cleavage in vitro[24]. The failure of GUARD RNAs to protect off-target sites could therefore be due to these rapid kinetics. In line with this, a 30-min pre-incubation of Cas9 and a 14-nt non-competitive GUARD RNA with the DNA substrate resulted in robust protection of the off-target DNA, even at a 1:1 ratio of GUARD RNA to gRNA (Fig. 2d). Further increasing the ratio of GUARD RNA to a five-fold excess led to complete protection from cleavage. In addition, on-target cleavage could be completely prevented using a competitive 14-nt GUARD RNA against *VEGFA*, demonstrating that GUARD RNAs can compete against perfectly complementary gRNAs (Fig. 2d). On-target protection of *VEGFA* required a higher ratio of GUARD RNA to gRNA, presumably due to the higher binding affinity of the on-target gRNA.

To better understand GUARD RNA positioning rules, we designed an in vitro Cas9 DNA cleavage assay for the *VEGFA* off-target site *CAVIN4*, whereby the same non-competitive GUARD RNA binding site was positioned incrementally further away within a synthetic DNA fragment (Supplementary Fig. 1). After incubation with Cas9 complexes, the remaining uncleaved DNA was quantified by qPCR. In this way, we could determine the optimal distance between the GUARD RNA and the off-target region for protection from Cas9 nuclease. As expected, the 14-nt non-competitive GUARD RNA against the *CAVIN4* off-target site provided significant protection when it was overlapping the off-target site, but also proved similarly effective when placed 10 bp away from the gRNA PAM (Fig. 2e). However, GUARD RNAs positioned 25 and 50 bp distal to the off-target gRNA PAM on either the 5' or 3' flank provided no protection (Fig. 2e), implying

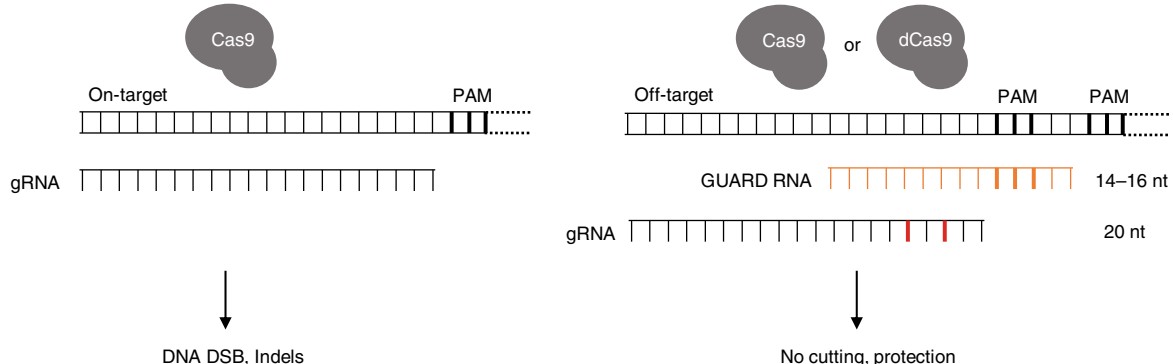

**Fig. 1 CRISPR GUARD protection of Cas9 off-target sites.** Schematic showing how short gRNAs (orange) forming catalytically inactive complexes with Cas9 can occupy specific off-target sites in the genome and compete with the mismatched nuclease competent gRNA (black), thereby providing protection from Cas9-mediated DNA cleavage. Mismatches in the gRNA are shown in red. RNA bases overlapping the gRNA PAM are shown in bold.

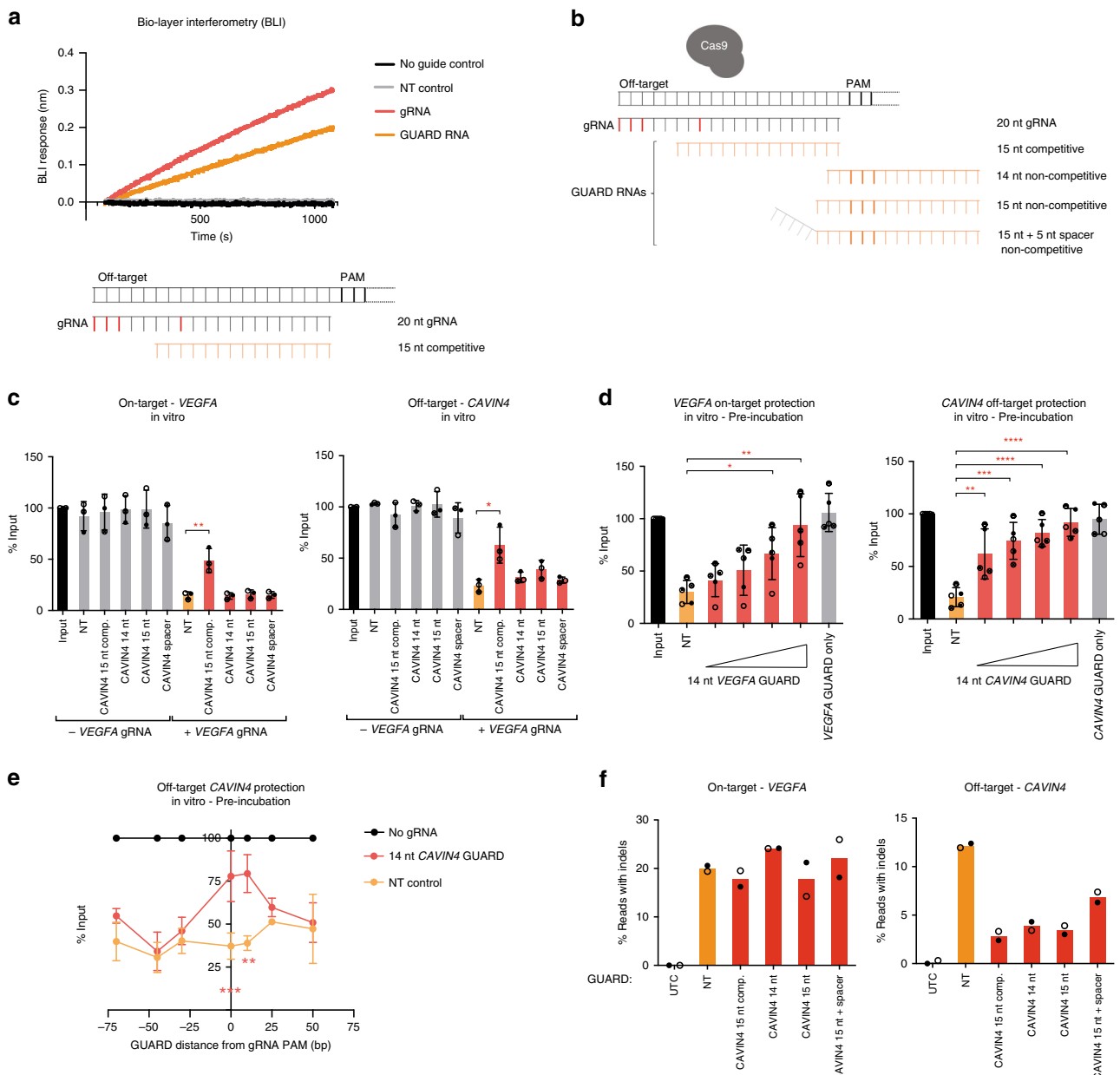

**Fig. 2 GUARD RNAs reduce Cas9 off-target cleavage activity without affecting on-target editing. a** Bio-layer interferometry (BLI) analysis of binding kinetics for dead Cas9 ribonucleoprotein complex to an immobilised biotinylated DNA substrate. Cas9 was precomplexed with tracrRNA and (NT) non-targeting control gRNA; (gRNA) 20-nt *VEGFA* gRNA with 4 mismatches; (GUARD RNA) a 15-nt GUARD RNA targeting the off-target site or no guide RNA. **b** CRISPR GUARD design. Schematic showing GUARD RNAs designs. Variation in length (14-nt versus 15-nt) and protospacer positioning likely influence binding energy and competition with the on-target gRNA (i.e., competitive/overlapping, non-competitive/proximal). Mismatches in the gRNA are shown in red. RNA bases overlapping the gRNA PAM are shown in bold. **c** Competition between on-target gRNA and GUARD RNA can be reduced by proximal positioning. On-target (*VEGFA*) and off-target (*CAVIN4*) DNA abundance was measured by qPCR following an in vitro Cas9 DNA cleavage assay. The corresponding GUARD RNA was added at a molar ratio of 5:1 to the *VEGFA* on-target gRNA (100 nM to 20 nM). **d** GUARD RNAs are more effective at blocking Cas9-mediated DNA cutting with pre-incubation in vitro. On-target (*VEGFA*) and off-target (*CAVIN4*) DNA abundance was measured by qPCR following an in vitro Cas9 DNA cleavage assay, with a 30 min pre-incubation with GUARD RNAs and Cas9, before addition of the on-target gRNA and Cas9. Increasing molar ratios of GUARD RNA to the *VEGFA* on-target gRNA were used (10:10, 20:10, 50:10, 100:10 nM). **e** GUARD RNAs are effective in blocking Cas9-mediated DNA cleavage of off-target sites within a 10 bp window of the gRNA PAM. Synthetic DNA fragments containing the *VEGFA* gRNA off-target site *CAVIN4* positioned progressively further away from the GUARD RNA binding site were quantified by qPCR following an in vitro Cas9 DNA cleavage assay (50:10 nM GUARD RNA to gRNA ratio). **f** CRISPR GUARD is effective at blocking off-target editing in cells. Indel rates from NGS of amplicons from Cas9-expressing HEK293 cells transfected with *VEGFA* gRNA and multiple GUARD RNA designs for protection of the *CAVIN4* proximal off-target site (25:25 nM GUARD RNA to gRNA ratio). Data represent the mean of two (**f**) independent experiments, or the mean ± SD of three (**c, e**) or five (**d**) independent experiments with symbols representing each replicate. For **a**, data are representative of two independent experiments. NT non-targeting gRNA, UTC untransfected control. Unpaired, two-tailed student's *t*-test. For **c**, *P = 0.0208, **P = 0.0083. For **d**, *P = 0.0164, **P = 0.0021 (*VEGFA*) and **P = 0.007 (*CAVIN4*), ***P = 0.0003, ****P < 0.0001. For **e**, **P = 0.0041 and ***P = 0.0001. Source data are provided as a Source Data file.

that GUARD RNAs are effective in rendering an off-target region inaccessible for editing if positioned ≤10 bp away from the gRNA PAM.

**Reduced off-target editing without affecting on-target editing**. Next, we tested the cellular activity of CRISPR GUARD. Using a Cas9-expressing HEK293 cell line[25], we co-transfected a *VEGFA* gRNA and multiple GUARD RNA designs (Fig. 2b) at equimolar ratios for protection of the *CAVIN4* proximal off-target site and performed next-generation sequencing (NGS) of amplicons to detect insertion and deletions (indels). Strikingly, all tested GUARD RNA designs significantly protected the off-target site (Fig. 2f). Notably, none of the GUARD RNA designs significantly interfered with on-target cutting efficiency, including the competitive GUARD RNA. Furthermore, unlike the in vitro scenario, phased delivery of the GUARD RNA before the gRNA provided no additional benefit (Supplementary Fig. 2). We reasoned that, in contrast to the in vitro setting, the kinetics of Cas9 binding to the off-target region is slower due to scanning of the mammalian genome, thus concomitant delivery of GUARD RNA is sufficient to allow for effective competition at off-target sites and may also reduce the effects of direct competition at the on-target locus with competitive GUARD RNAs. Due to the apparent efficacy of the 14-nt and 15-nt GUARDs in vitro and in cells, we disregarded the 15-nt GUARD RNA with a 5-nt mismatched spacer sequence (Fig. 2b), as this provided the least protection (Fig. 2f) and could potentially give rise to a catalytically active Cas9 complex if the GUARD RNA binds to a DNA site where the mismatched spacer has >2-nt of base pairing.

Next, we investigated the optimal dosing for CRISPR GUARD. We co-transfected a gRNA against *VEGFA* and increasing concentrations of GUARD RNA to protect the *CAVIN4* proximal off-target site. Although an equimolar ratio of GUARD RNA to gRNA was effective in reducing off-target indel rates (Fig. 2f), higher concentrations of GUARD RNA provided additional reduction of off-target editing (Fig. 3a). Notably, the highest concentration of GUARD RNA (10:1 ratio) reduced off-target indel rates to background levels (Fig. 3a). We noted that indel rates in untransfected control cells for the *CAVIN4* proximal region are approximately 0.3%, owing to intrinsic error in the NGS of a polymeric cytosine tract at this locus. We assume that Cas9 concentration is in excess in the cell as increasing the ratio of GUARD RNA to gRNA did not have a negative impact on on-target editing, so in this context we recommend routinely adopting a 5:1 ratio of GUARD RNA to gRNA.

To determine if CRISPR GUARD was also effective in the therapeutically relevant setting of Cas9 ribonucleoprotein (RNP) delivery, we separately precomplexed catalytically inactive Cas9 (dCas9) protein with a GUARD RNA against *VEGFA* off-target site *TENT4A*, and wild-type Cas9 protein with the *VEGFA* on-target guide RNA, to co-deliver these RNPs for protection and cutting, respectively. In principle, it is an option to use a full-length 20-nt gRNA for protection with dCas9. However, we continued to use 14-nt GUARD RNAs in this scenario in order to prevent guide swapping in the cell[26], whereby wild-type Cas9 acquires full-length gRNA against the off-target site. Encouragingly, CRISPR GUARD using RNPs successfully protected *VEGFA* off-target *TENT4A*, and there was no detectable impact of protection of the *TENT4A* off-target site on the other known off-target sites by redistribution of active Cas9 (Supplementary Fig. 3).

To begin to understand GUARD RNA design rules, we assessed the activity of an array of GUARD RNAs protecting known endogenous off-target sites of *EMX1* and *VEGFA* gRNAs[21–23,27], assessing nucleotide composition and positioning.

We used equimolar GUARD RNA and gRNA in order discern off-target protection that would be masked at higher GUARD RNA concentrations (Fig. 3a). Notably, both *MYC* and *CAVIN4* GUARDs worked exceptionally well, reducing indel rates from 2.6 ± 0.3% to 0.08 ± 0.08%, and 19 ± 1.9% to 3.3 ± 0.4%, respectively (Fig. 3b). All GUARD RNAs tested reduced off-target indel rates except for one (*VEGFA* off-target *HDLBP*). Upon inspection of the non-functional *HDLBP* GUARD RNA, we noted that it had relatively low GC-content compared to the cognate on-target gRNA, potentially leading to low binding affinity and reduced ability to compete at the off-target locus. Introduction of a revised GUARD RNA with higher GC-content, length and increased overlap of the gRNA seed region, achieved only modestly improved protection (Supplementary Fig. 4). We only observed protection at a 5:1 molar ratio of GUARD RNA to gRNA, and no protection at a 1:1 ratio. Taken together, these data suggest that some off-target loci are more amenable to protection by CRISPR GUARD than others, indicating more systematic investigation is warranted. This is likely contingent on the relative affinities of the GUARD RNA and the cognate gRNA for the off-target region.

As protection of one off-target with CRISPR GUARD apparently does not increase editing at other off-target sites (Supplementary Fig. 3), we attempted multiplexing of three GUARD RNAs together to assess if it was possible to protect multiple off-target sites simultaneously. We performed multiplexed GUARD RNA experiments by transfecting Cas9 expressing HEK293 cells with on-target gRNA and three independent GUARD RNAs all at equimolar concentrations. For both *EMX1* and *VEGFA*, multiplex GUARD RNA delivery was generally effective in protecting multiple off-target sites from indel formation (Supplementary Fig. 5). Consistent with the in vitro data (Fig. 2b), the non-competitive GUARD RNAs used in these experiments showed no effect at the on-target site (Supplementary Fig. 5), thus allowing robust protection of several off-targets whilst maintaining efficient on-target editing.

To further validate the safety of CRISPR GUARD, we transfected various functional GUARDs (14-nt and 15-nt in length) at the highest effective concentration used in this study (125 nM) and assessed if they could support Cas9-mediated DNA cleavage in cells by deep sequencing of amplicons. In concordance with our in vitro data (Fig. 2b, c), none of the GUARD RNAs could generate Cas9-induced indels in isolation, in contrast to a 20-nt control gRNA at the same concentration (~82% indel formation), affirming that GUARD RNAs form nuclease-dead complexes with Cas9 in the cell (Supplementary Table 1).

**CRISPR GUARD for base editing**. Next, we applied CRISPR GUARD to base editing. Due to the strict positioning requirements for base editing activity, gRNA design possibilities are more limited and therefore off-targets are harder to avoid[28,29]. We reasoned that CRISPR GUARD could reduce Base Editor 3 (BE3) activity at off-target sites, since the activity window of BE3 is mostly absent from GUARD RNAs (Supplementary Fig. 6a), and optimal activity of BE3 is dependent on nickase Cas9 nuclease acitivity[28], which is compromised with short gRNAs[18–20]. Using a HEK293 cell line expressing BE3, we tested GUARD RNA designs for protection of *EMX1* and *VEGFA* off-target sites (Supplementary Fig. 6b). We demonstrated significant protection of the *EMX1* off-target cytosines proximal to *MYC* (a reduction from 6.03 ± 0.6% to 0.98 ± 0.09% editing), and the *VEGFA* off-target cytosines proximal to *CAVIN4* (a reduction of 23.33 ± 0.45% to 11.37 ± 2.72%; Fig. 4a). These GUARD RNAs also performed well in the CRISPR/Cas9 system (Fig. 3b). As with Cas9, introduction of GUARD RNAs did not compromise on-target editing efficiencies (Fig. 4a).

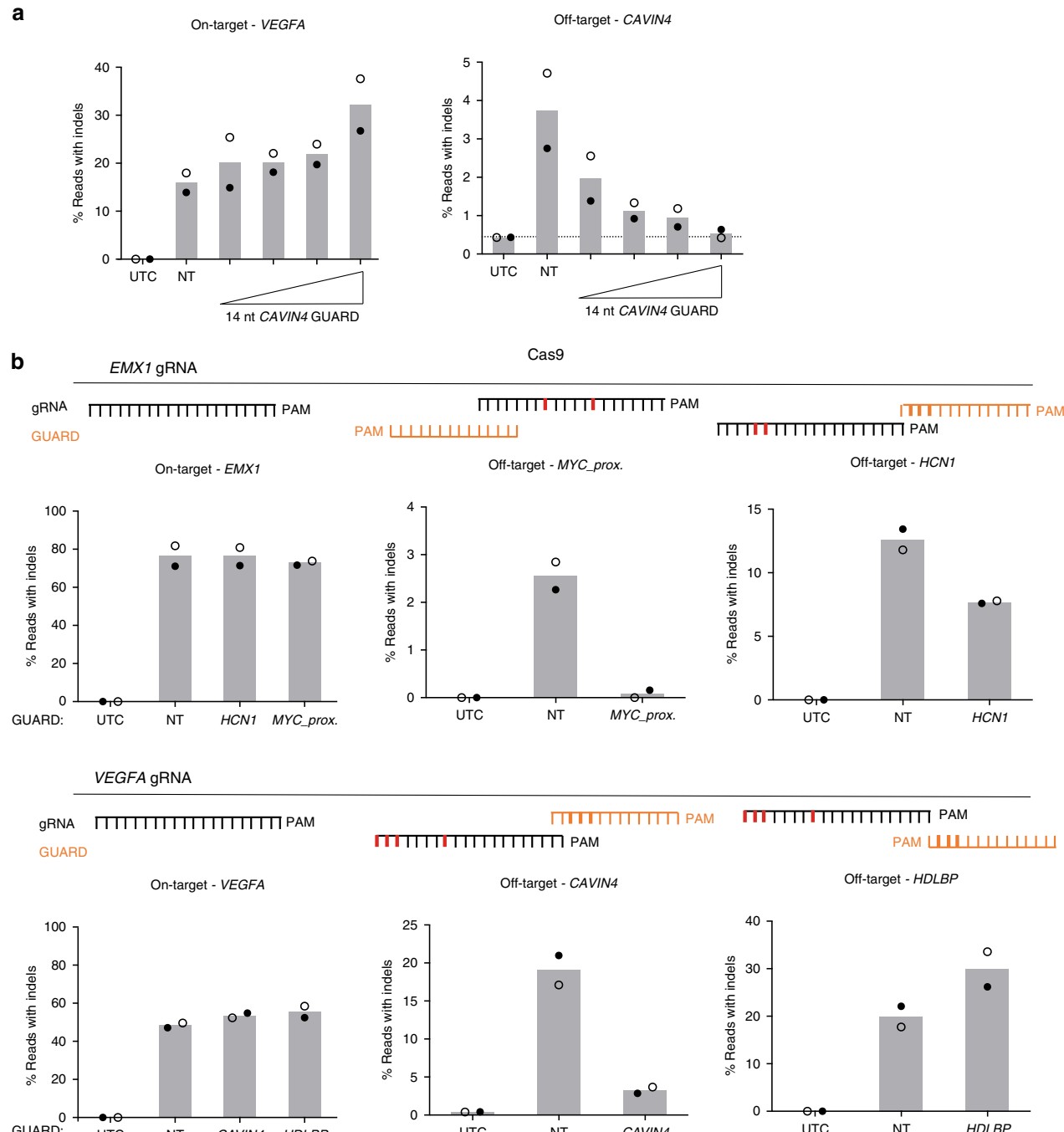

**Fig. 3 Protection of multiple, endogenous off-target sites with CRISPR GUARD. a** Increasing GUARD RNA concentration leads to complete protection of Cas9 off-target sites in cells. Indel rates from NGS of amplicons from Cas9-expressing HEK293 cells transfected with *VEGFA* gRNA and increasing concentrations of 14-nt GUARD RNA for protection of the *CAVIN4* off-target site (10:10, 20:10, 50:10, 100:10 nM). The total concentration of delivered RNA in each case was kept constant by co-delivery of NT gRNA (first condition is 100 nM NT). **b** Indel rates from amplicons sequencing of Cas9-expressing HEK293 cells transfected with *EMX1* or *VEGFA* on-target gRNAs, and a single GUARD RNA (25:25 nM ratio). GUARD RNA positioning relative to the off-target protospacer sequence is shown schematically in each panel. Data represent the mean of two independent experiments with symbols representing each replicate. Source data are provided as a Source Data file.

To date, the activity of short gRNAs for base editing applications has not been systematically investigated. The *HDBLP* and *MFAP1* GUARD RNAs are predicted to expose cytosines as single-stranded DNA near the BE3 deaminase activity window (Supplementary Fig. 6a and 6b). Interestingly, when BE3 expressing cells were transfected with both on-target gRNA and GUARD RNA, an overall increase in deamination rates was observed (Fig. 4b). Strikingly, the *HDBLP* GUARD RNA

introduced two distinct G-to-A mutations that were not detected in the absence of GUARD RNA (Fig. 4b). These mutations were found both linked and unlinked to those generated by the on-target gRNA, suggesting they could be occurring in the same editing event (Supplementary Fig. 6c). Moreover, when we transfected the *MFAP1* GUARD RNA alone (without *EMX1* on-target gRNA), we could detect a low but significant number of C-to-T deamination events (Fig. 4c; 3.44 ± 0.40%), suggesting that

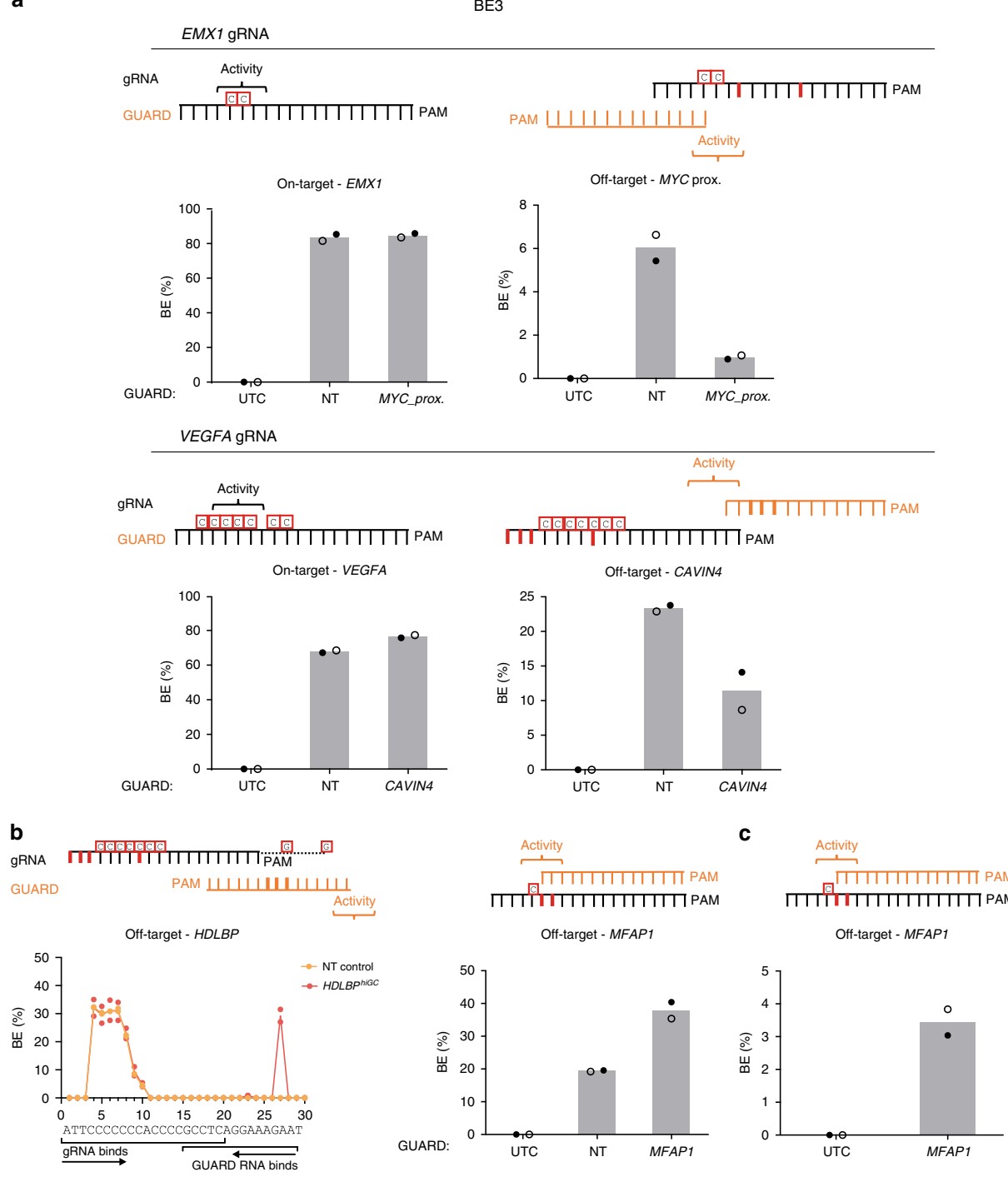

**Fig. 4 GUARD RNAs can reduce off-target base editing if designed to avoid cytosine exposure. a** Protection from off-target cytosine deamination with CRISPR GUARD. Base editing rates from amplicon sequencing of BE3-expressing HEK293 cells transfected with *EMX1* or *VEGFA* on-target gRNA (25 nM) and the indicated GUARD RNA (125 nM). Shown is the cumulative editing rate for the cytosines highlighted. The predicted cytosine deamination activity window is indicated. **b** Increased base editing frequencies with GUARD RNAs that expose cytosines. Base editing rates from amplicon sequencing of BE3-expressing HEK293 cells transfected with *EMX1* (left) or *VEGFA* (right) on-target gRNA (25 nM) and the indicated GUARD RNA (125 nM). **c** GUARD RNAs can facilitate base editing without full-length on-target gRNAs. Base editing rates from amplicon sequencing of BE3-expressing HEK293 cells transfected with *MFAP1* GUARD RNA only (125 nM). Data represent the mean of two independent experiments with symbols representing each replicate. Source data are provided as a Source Data file.

short GUARD RNAs are sufficient to independently support base editing in cells. It is likely that the low level of base editing observed with short GUARD RNAs alone is due to the lack of nickase activity, which is analogous to editing with base editor 2 versions[28].

**A high-throughput screen identifies functional GUARD RNAs.** Thus far, we have analysed the performance of a relatively small number of GUARD RNAs targeting endogenous loci. To systematically screen all possible Cas9 or BE3 GUARD RNAs for a given target, we adapted a high-throughput approach[30,31] using a

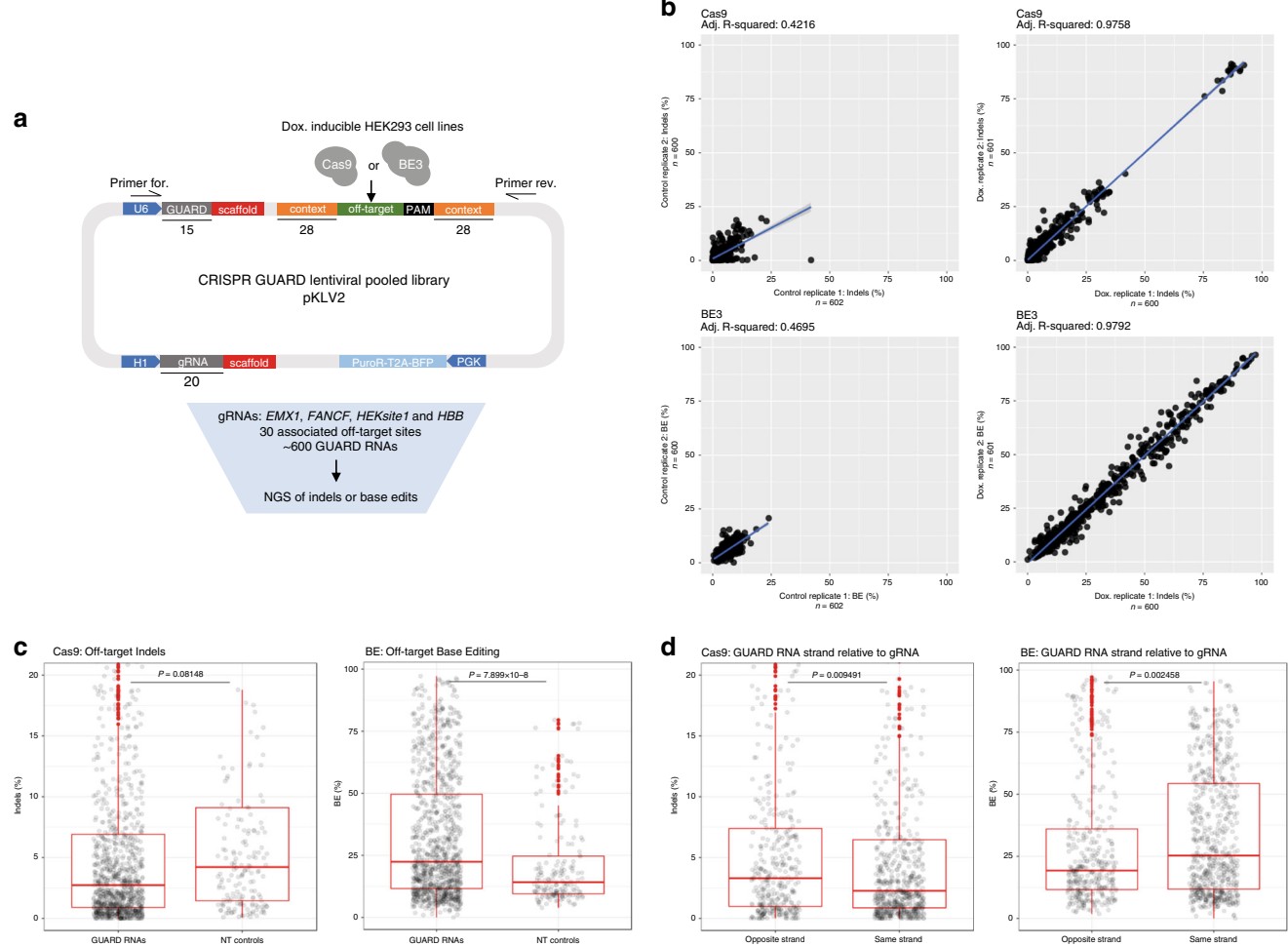

**Fig. 5 High-throughput screening identifies functional GUARD RNAs. a** Schematic of the lentiviral pooled screening approach to analyse off-target editing frequencies in cells expressing different GUARD RNA species. Numbers indicate the length of the plasmid segment in base pairs. **b** Scatter plots of indels (Cas9) or SNPs (base edits) for two independent replicate experiments with or without induction of Cas9 or BE3 expression with doxycycline for 48 h. **c** Box and whiskers plot of off-target indel or base editing frequency for coding and non-targeting (NT) GUARD RNAs in the Cas9 or base editor screen, respectively. NB: some outliers fall outside of the y-axis limit. **d** GUARD RNAs binding to the same DNA strand as the gRNA are more effective at blocking Cas9 off-target editing but can increase base editing efficiency. Box and whiskers plot of off-target indel or SNP frequencies in the screens. NB: some outliers fall outside of the y-axis limit. Box and whiskers plot: centre line, median; box limits, upper and lower quartiles; whiskers, 1.5× interquartile range; points, outliers. Data were compared using unpaired, two-tailed student's t-tests.

library of lentiviral constructs that express both gRNA and GUARD RNA and harbour the off-target sequence within 79 bp of its genomic context. Using this system, we screened the performance of ~600 GUARD RNAs including NT GUARD RNA controls, obviating amplification of individual endogenous loci to analyse editing events (Fig. 5a). We tested four gRNAs based on their therapeutic relevance (*HBB;* a therapeutic target for correcting sickle-cell disease[14]) or because of an extensive knowledge of experimentally validated off-target sites (*EMX1, FANCF,* and *HEKsite1*). Each gRNA expression vector was linked to one of 30 experimentally validated off-target sites[2]. For each of the off-target sites, we tested all possible 15-nt GUARD RNAs within a 10 bp window either side of the mismatched gRNA protospacer with an NRG PAM (where R is a purine). We introduced the lentiviral pooled library into HEK293 cells expressing doxycycline-inducible Cas9 or BE3 and measured the number of indels or base edits for each GUARD RNA by NGS (Fig. 5a).

As expected, we observed a significant induction of editing at off-target sites by Cas9 and BE3 in the presence of doxycycline with a high correlation of editing frequencies between biological replicates (Fig. 5b and Supplementary Fig. 7). Editing profiles were compared

to those derived from deep sequencing of the original plasmid library, such that we focussed on mutations caused by Cas9 or BE3 expression in the cell in downstream analyses. Low frequency editing was observed without doxycycline, likely reflecting a degree of leakiness in the inducible system. To further validate the screen, we compared the Cas9-driven indel rates in the screen with those detected with GUIDE-seq methodologies on endogenous genomic loci[2] and found that the observed mutation rates correlated well (Supplementary Fig. 8), verifying the physiological relevance of the system[30,31].

Comparing the abundance of Cas9-driven off-target indels, we observed that expression of GUARD RNAs tended to reduce off-target mutations compared to cells expressing NT GUARD molecules (Fig. 5c). It was also clear that many of the screened GUARD RNAs have minimal effects, which is perhaps to be expected as we did not pre-select GUARD RNAs based on sequence features and many of these GUARD molecules may perform better at higher concentrations relative to the gRNA (Fig. 3a).

Conversely, we discovered that GUARD RNAs significantly increased base editing at off-target sites on average when compared to NT GUARD RNA controls (Fig. 5c), consistent

with some GUARD RNAs being able to support base editing (Fig. 4). BE3 had a higher rate of off-target editing than Cas9 (Fig. 5c), with most SNPs detected being C->T and G->A mutations, consistent with cytosine deamination (Supplementary Fig. 9). Notably, base editing with GUARD RNAs was promoted by binding to the same DNA strand as the gRNA (Fig. 5d), presumably because the full-length gRNA-BE3 complex nicks the opposite strand, thus encouraging repair of the unedited strand and retention of the mutation[28]. In contrast, Cas9 GUARD RNAs binding on the same DNA strand as the gRNA significantly reduced indel rates (Fig. 5d), implying that this may be sterically more obstructive.

**Sequence features of functional GUARD RNAs**. To delineate features of protective GUARD RNAs, we designated GUARD RNAs that reduced off-target editing by more than two standard deviations from the mean of the NT GUARD RNA controls as functional molecules (Figs. 6a, 6b and Supplementary Fig. 10). We selected a pair of reproducibly functional and non-functional Cas9 GUARD RNAs and performed validation experiments on the endogenous genomic locus. Results from these experiments were in direct concordance with the screening data supporting the validity of this categorisation (Fig. 6c).

Half of the analysed off-target sites (15 of 30) had functional GUARD RNAs for Cas9, and 80% (24 of 30) had functional GUARD RNAs for BE3 (Supplementary Fig. 10). Functional GUARD RNAs were evenly distributed between off-targets with low and high mutation rates and for off-targets with different numbers of gRNA mismatches (Fig. 6a). Of the 510 targeting GUARD RNA designs tested, $18 \pm 1\%$ showed functional protection from Cas9 off-target activity, with $23 \pm 0.3\%$ of these reducing off-target editing to below 0.5%. For BE3, fewer GUARD RNAs showed functionality, with $8 \pm 1\%$ showing reproducible protection of off-target sites (Fig. 6b). Only one GUARD RNA was able to reduce BE3 editing to below 0.5%, consistent with the finding that some GUARD RNAs can also support base editing. Specifically, the presence of a cytosine within the first 2-nt of the 15-nt GUARD RNA gave rise to significantly more base editing events (Fig. 6d). Concordant with GUARD RNAs with high affinity having superior protective effects, NGG PAMs were significantly enriched in functional GUARD RNAs over NAG (Fig. 6e). Selecting only Cas9 GUARD RNAs with an NGG PAM significantly increased the percentage of functional Cas9 GUARD RNAs to $26 \pm 1\%$. Moreover, the proportion of functional GUARD RNAs tended to increase with higher GC-content (Supplementary Fig. 11a). Finally, GUARD RNAs with a higher degree of spatial overlap with the gRNA, especially at the seed and PAM region, led to superior off-target protection (Supplementary Fig. 11b). Taken together, these data support a model of off-target protection by CRISPR GUARD through direct competition with the mismatched gRNA and highlight important parameters for GUARD RNA design (Fig. 6f).

Finally, we generated a publicly available tool for automated GUARD RNA design called CRISPR GUARD Finder (https://www.sanger.ac.uk/tool/crispr-guard-finder/ and https://github.com/MatthewACoelho/CRISPRGUARDFinder), which predicts potential off-target sites for a given gRNA, and generates a list of possible GUARD RNAs with relevant sequence features. An example is given in Supplementary Fig. 12.

## Discussion

In this report, we reveal CRISPR GUARD as a rapidly implementable tool to reduce off-target editing by Cas9 without reducing on-target editing efficiency. Importantly, on-target editing is maintained even when GUARD RNAs are multiplexed to protect multiple off-target sites simultaneously (Supplementary Fig. 5). Using

in vitro assays, in-cell assays and screening hundreds of GUARD RNAs, we show that the majority of off-target sites can be protected by GUARD RNA molecules and identify key parameters for GUARD RNA design for Cas9 and BE, including distance from the off-target, GC-content, PAM, DNA strand, concentration and cytosine positioning (Fig. 6f). In addition, GUARD RNAs with extensive homology to the on-target site may interfere with on-target editing in some cases (Fig. 2c).

GUARD RNA design rules for BE is complicated by the ability of short gRNAs to direct cytosine deamination in isolation. Many features of Cas9 GUARD RNA design were not significantly enriched in functional GUARD RNAs for base editing, presumably because features promoting strong GUARD RNA binding also promote base editing. This property could be exploited to reduce bystander cytosine editing at the 5' of the gRNA, or may be used in tandem with full-length gRNAs to broaden the editing window when multiple editing events are desired. Nevertheless, as long as cytosines are avoided within the base editing window, functional BE GUARD RNAs can be identified that significantly reduce off-target base editing (Fig. 6b).

Although the minority of GUARD RNAs were protective in the Cas9 lentiviral screen where we cannot directly control the molar ratio between gRNA and GUARD RNA, it is clear that higher doses of GUARD RNAs are often required to compete with the full-length gRNA. Moreover, GUARD RNAs that satisfy sequence features identified here are more likely to be successful in blocking off-target mutation. At high effective concentrations, we verify that GUARD RNAs cannot support nuclease activity in vitro or in cells. However, it is possible that GUARD RNA-Cas9 complexes might transiently interfere with transcription[32]. Thus, more work is required to refine GUARD RNA design computationally to minimise the number of potential binding sites within the genome (intrinsically high for short polynucleotides), and to optimise binding energies relative to the cognate mismatched gRNA. For example, locked nucleic acid bases could further improve the specificity and binding energy of GUARD RNAs to favour displacement of the mismatched gRNA. In addition, the CRISPR GUARD Finder tool will be useful in allowing researchers to rapidly assess the optimal CRISPR GUARD designs for gene editing experiments. Despite our demonstration that multiplexing is feasible for CRISPR GUARD, there may be instances where a particular gRNA is predicted to have many off-target sites and screening all promising GUARD RNA designs may not be feasible. In such situations, we suggest that the user considers refining gRNA design, and failing this, prioritising off-targets that are experimentally validated, bioinformatically scored as most probable, or likely to be particularly detrimental (e.g., genic).

We envisage that CRISPR GUARD could be employed for therapeutic applications that require high editing efficiency and where high-fidelity Cas9 variants cannot fully supress persistent off-target editing[14]. We demonstrate CRISPR GUARD is effective with RNP delivery, where editing efficiencies may be maximised by complexing GUARD RNAs with dCas9, and the gRNA with active Cas9. However, in situations where this becomes less practical (e.g., multiplexing GUARD RNAs), co-delivery with active Cas9 will be more appropriate. Many of our experiments involved co-transfection of gRNA and GUARD RNAs in Cas9-expressing cells lines, with no apparent consequences of guide-swapping[26]. The design principles of CRISPR GUARD could be easily adapted for use with Cas9 orthologues, TALENs and zinc-finger nucleases, while Cas9 variants that utilise distinct PAMs will increase the number of suitable GUARD RNAs for each off-target. By demonstrating that we can also block editing at perfectly complementary on-target sites (Fig. 2d), we open the possibility for controlling the kinetics of gene editing by introducing GUARD RNAs as an off-switch. In summary, CRISPR GUARD

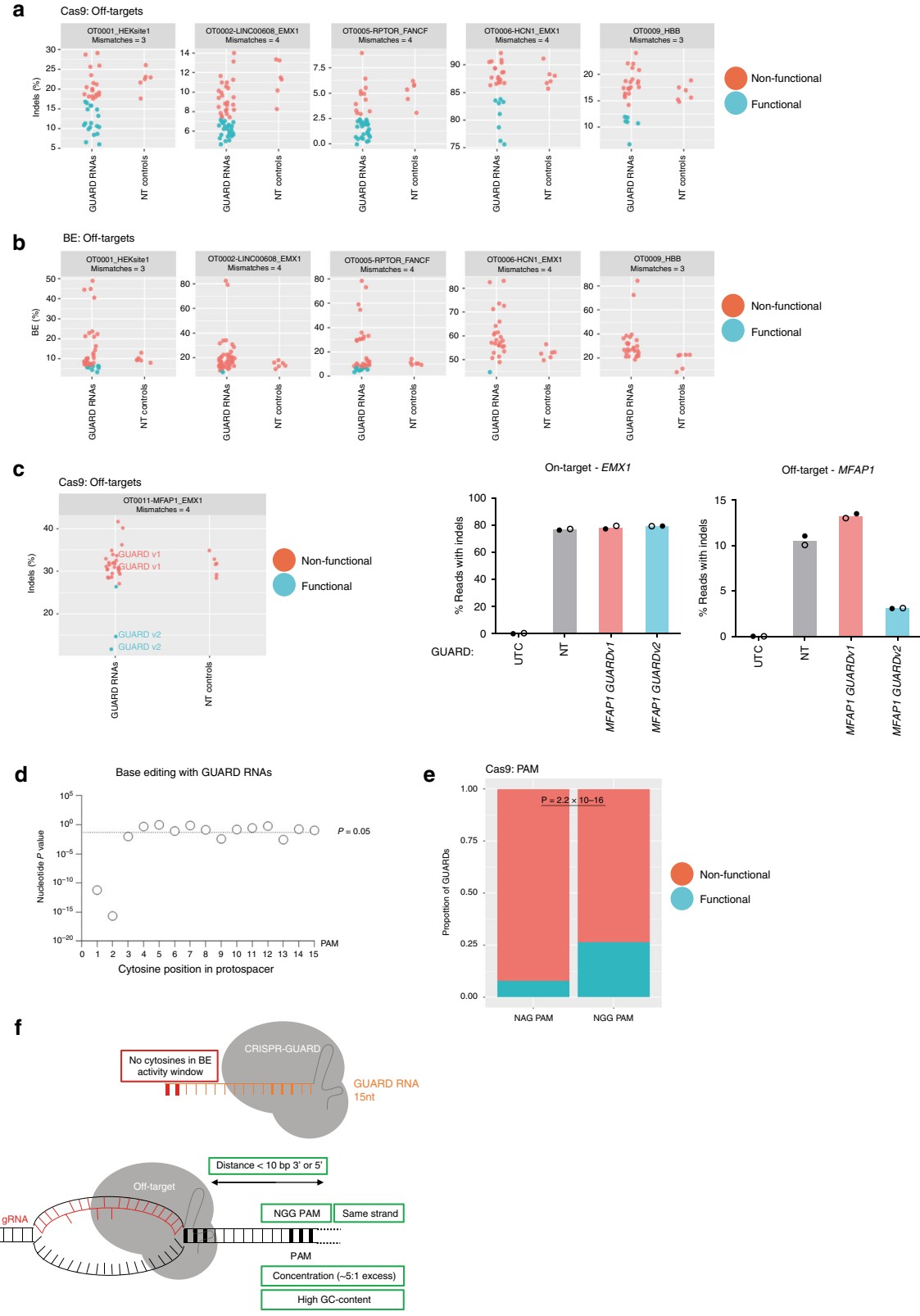

has exciting potential for improving the specificity and safety of genome editing. During review, a similar technology report was published with complementary and supportive data[33].

## Methods

**CRISPR-GUARD RNAs**. GUARD RNA and full-length gRNA sequences are listed in Supplementary Table 2 and are AltR modified IDT crRNAs.

**Bio-layer interferometry (BLI)**. DNA substrate was generated from duplexing a biotin-conjugated oligo with an unlabelled oligo coding for the *CAVIN4* off-target site of the *VEGFA* gRNA. Labelled strand (AGCCACAACCCTGTTGGACGTCC TGAGGCGGGGTGGGGGGGTGTGCAAGGGAACTCTCC), unlabelled strand (GGAGAGTTCCCTTGCACACCCCCCCACCCCGCCTCAGGACGTCCAACAG GGTTGTGGCT). GUARD RNA (CCCCCACCCCGCCTC), gRNA (GACCCCC TCCACCCCGCCTC). BLI measurements were performed at 25 °C using an Octet Red96 instrument (ForteBio). The measurement buffer consisted of PBS

**Fig. 6 GUARD RNA features that significantly reduce off-target editing. a** GUARD RNA performance for Cas9 and, (**b**) BE3. Base editing rates for BE (% of NGS reads with SNPs) or indel rates for Cas9 (% of NGS reads with indels) for a subset of off-target sites. Off-targets are labelled with an identifier, gene name (if genic), gRNA name, and the number of mismatches between the gRNA and the off-target site. Data are pooled from two independent experiments from cells treated with doxycycline. Each point represents a GUARD RNA and are compared to three non-targeting (NT) control RNAs. GUARD RNAs are classed as functional if they reduce off-target editing by at least two standard deviations from the mean of the NT controls. Data shown here is a subset of that presented in Supplementary Fig. 10. **c** Lentiviral plasmid screening reveals GUARD RNAs that are effective at endogenous genomic loci. (Left panel) indel rates from GUARD RNA screening of *EMX1* gRNA off-target site *MFAP1*, highlighting non-functional GUARD RNA (v1) and functional GUARD RNA (v2). Screening data shown here is a subset of that presented in Supplementary Fig. 10. (Right panels) indel rates from amplicon sequencing of on-target and off-target endogenous loci from Cas9-expressing HEK293 cells transfected with *EMX1* gRNA and GUARD RNA v1, GUARD RNA v2 or a NT control (all at 25 nM). Data represent mean of two independent experiments with symbols representing each replicate. **d** GUARD RNAs can mediate base editing at position 1 and 2 within the 15-nt GUARD RNA. GUARD RNAs were grouped according to whether they had a cytosine at each nucleotide position or not, and *P* values were generated by comparing the % of NGS reads with SNPs between GUARD RNA groups with an unpaired, two-tailed student's *t*-tests. **e** Functional GUARD RNAs predominantly have NGG PAMs. The proportion of functional GUARD RNAs from the Cas9 screen that had NGG PAMs versus NAG PAMs was compared using a two-sided Fisher's exact test. **f** Model depicting important parameters for GUARD RNA design. For GUARD RNAs that reduce Cas9 off-target editing, reducing distance from the gRNA, having an NGG PAM, binding to the same DNA strand as the gRNA, increased GC-content and high concentration in the cell will all increase the effectiveness of CRISPR GUARD. For BE3, the presence of cytosines at position 1 or 2 is to be avoided for 15-nt GUARD RNAs to prevent editing. Source data are provided as a Source Data file.

supplemented with 0.005% Tween-20, 1 mM MgCl$_2$ and 50 μg ml$^{-1}$ heparin. SA tips were used to immobilise ca. 0.1 unit (nm) of 5′-biotinylated duplex DNA oligo. Subsequently, tips were typically dipped in the measurement buffer for 30 s and transferred to precomplexed dCas9:tracrRNA:guideRNA (formed at 100 nM:200 nM:200 nM, respectively) for a 1000 s association step.

**In vitro Cas9 cleavage assays**. Cas9 DNA cleavage reactions were made up in a final reaction volume of 10 μL and incubated at 37 °C for 1 h. Reactions consisted of: *S. pyrogenes* Cas9 (120 nM; NEB), 1× NEB Cas9 reaction buffer, on-target gRNA (10 nM), a range of concentrations of GUARD RNA (10–100 nM), 120 nM tracrRNA, a purified PCR amplicon containing the gRNA protospacer sequence (0.5 nM) or a purified PCR amplicon containing the mismatched off-target pro-tospacer sequence (0.5 nM), as indicated in the figures. In each case, the non-targeting gRNA (NT) was used at the highest concentration (100 nM) as a negative control and used to back-calculate the final concentration of GUARD RNA in the titration experiments such that the total concentration of RNA in each condition was equivalent. For control reactions with GUARD RNA only, GUARD RNA was used at the highest concentration (100 nM). For pre-incubation with GUARD RNAs, the reactions were set up as above, except for a 30 min preincubation of Cas9 (120 nM in 5 μL) with GUARD RNA and the PCR amplicon, before addition of the gRNA and Cas9 (120 nM in 5 μL) for a further 1 h. For Fig. 1e, gblocks (IDT) were synthesised containing the GUARD RNA binding region in the following positions: overlapping, 10 bp away, 25 bp away or 50 bp away from the off-target protospacer. These gblocks were used as input in Cas9 DNA cleavage assays as above (0.5 nM input DNA). Reactions were stopped by the addition of an equal volume of 100 mM EDTA and heating for 5 min to 65 °C and subsequently diluted 10-fold with water before analysis of cutting efficiency by qPCR. The non-targeting guide RNA was also from IDT and has the protospacer sequence: GCCCCGCCGCCCTCCCCTCC[11].

**qPCR**. qPCR primers used are listed in Supplementary Table 3. An ABI 7900 (ThermoFisher) and Fast SYBR Green Master Mix (ThermoFisher) were used to quantify cutting efficiency using the no Cas9 condition as a control for input, where % input = $100^2$(Ct input − Ct experimental).

**CRISPR GUARD in cells**. HEK293 (ATCC) Cas9-expressing cells and BE3-expressing cells were cultured in RPMI medium (Sigma-Aldrich) supplemented with 10% FCS (ThermoFisher), 1% GlutaMAX (ThermoFisher) in a 37 °C, 5% CO2, 95% air incubator. HEK293 Cas9-expressing cells and BE3-expressing cells were generated using ObLiGaRe-mediated incorporation of an expression cassette into the AAVS1 safe-harbour locus by co-transfection of plasmids encoding AAVS1-targeting zinc-finger nucleases and the expression plasmid[25,34], followed by selection in G418. Parental HEK293 cells were acquired from ATCC and all cell lines used were STR profiled and verified as mycoplasma-free.

For CRISPR GUARD experiments in cells, 200,000 cells were seeded in a 24 well tissue culture plate in 500 μL medium. The following day, cells were co-transfected. Briefly, each gRNA (crRNA AltR modified; IDT) was separately precomplexed with tracrRNA (IDT) at a 1:1 molar ratio in IDT duplex buffer by heating to 95 °C for 5 min. After duplexing with tracrRNA, on-target gRNA, non-targeting gRNA, or GUARD RNAs were diluted in Opti-MEM (ThermoFisher) and co-transfected with RNAiMAX (ThermoFisher) according to manufacturer's instructions. Forty-eight hours later, DNA was extracted using DNeasy Blood and Tissue kit (Qiagen) and used for amplicon sequencing.

For RNP experiments, Cas9 RNPs were delivered using the Neon electro-poration system (ThermoFisher) using protocol 20. Briefly, 500,000 HEK293 cells

were electroporated with 1 μg of Cas9 precomplexed with gRNA and 1 μg of dCas9 precomplexed with GUARD RNA, with 0.1 μg of AltR electroporation enhancer (IDT).

**Amplicon sequencing and analysis**. Primers used for amplicon sequencing are listed in Supplementary Table 3. PCR reactions were performed using Phusion Flash Hi-Fidelity PCR Master Mix (ThermoFisher) using an optimised number of PCR cycles ranging from 22 to 25. The NGS indexing PCR was 10 cycles, using 1 ng of purified product from PCR1 as input. PCR fragments were purified using SPRI beads (MAGBIO), size verified using the QIAXcel (Qiagen), quantified with the Qubit (ThermoFisher) and finally pooled and sequenced on a NextSeq 500 or MiSeq (Illumina). Bioinformatics analysis of NGS data was performed essentially as described[21], using Fast Length Adjustment of Short reads (FLASH v1.2.11) for paired reads, BWA-MEM for alignment to the human genome, and Samtools to generate indexed BAM files and variant calling, with >0.001 allele frequency and >1000 read cut-off.

**CRISPR GUARD screening plasmid construction**. Using a scaffoldless version of the pKLV2 lentiviral base vector, we integrated a gBlock (IDT) containing H1 promoter-gRNA-scaffold into the Apa1-Mlu1 site using Gibson assembly (NEB). This produced four distinct plasmids containing either the *EMX1*, *FANCF*, *HEK-site1*, or *HBB* gRNA. We used the improved scaffold (encoding the tracrRNA) for gRNA and GUARD RNAs. Next, we designed all possible GUARD 15-nt sequences for each off-target site using our custom tool (https://www.sanger.ac.uk/tool/crispr-guard-finder/ and https://github.com/MatthewACoelho/CRISPRGUARDFinder) and designed an oligo library for each gRNA where each of the GUARD sequences are linked to the corresponding off-target embedded between 28 bp of flanking genomic context on either side. For each off-target sequence, we also included three non-targeting GUARD RNA controls, which were truncated versions of published Avana NT control guide RNAs[11], verified to have minimal complementarity to the human genome (sequences are listed in Supplementary Table 2). Single stranded oligo pools were obtained from IDT and were PCR amplified (10 PCR cycles) to make them double-stranded and to append adaptors containing short homology arms for Gibson assembly (NEB). Double-stranded oligo-pools were SPRI bead purified (MAGBIO) and subsequently inserted into the BbsI site of one of the four corresponding lentiviral plasmids to generate vectors expressing the gRNA and GUARD RNA. We pooled at least four transformations by electroporation of NEB Stable bacteria (NEB) to maintain library diversity on Amp agar plates and then harvested the DNA from a 3-h Amp liquid culture derived from a plate-scrape of colonies. Because the length of oligo synthesis was limiting, we designed the oligos to contain an internal AvrII site, which we then used to complete the tracrRNA scaffold sequence with a short PCR fragment using Gibson assembly (NEB). We harvested plasmid DNA as above and the resulting four completed libraries were combined into one pool in a ratio reflecting the total number of unique constructs in each sub-pool. The single pool of plasmids was sequence verified by NGS using primers designed to capture the GUARD and the off-target sequence.

**CRISPR GUARD screen execution**. We generated lentiviral particles in HEK293T cells with psPAX2 and pMD2.G packaging plasmids using FuGeneHD transfection reagent (Promega). We empirically determined the titre required to infect 40% of cells in the presence of 8 μg/ml Polybrene using BFP fluorescence detected by flow cytometry. The screen was performed in two independent infections treated as biological replicates for HEK293-Cas9 or HEK293-BE3 cells. The day after infection, HEK293 cells were selected for three days with puromycin (1 μg/ml), and surviving cells were induced with doxycycline (0.1 μg/ml) to induce

expression of Cas9 or BE3. Control cells were treated with medium alone. After 48 h, DNA was harvest and purified using DNeasy Blood and Tissue kit (Qiagen). To maintain sample complexity, we pooled $24 \times 20$ µl PCR reactions, each using 500 ng of genomic DNA as input. For PCR1 we used 22 cycles and for the subsequent indexing PCR2 we used 10 cycles with 5 ng of purified PCR1 as input.

**CRISPR GUARD screen analysis.** NGS was performed on a NextSeq or MiSeq (Illumina) and fastq files were grouped into pairs as above using Fast Length Adjustment of Short reads (FLASH v1.2.11). A custom Perl script (https://www.sanger.ac.uk/tool/crispr-guard-finder/ and https://github.com/MatthewACoelho/CRISPRGUARDFinder) was used to process the data. Firstly, we scanned for a GUARD sequence within the first 30 bp of the paired reads in the specific U6-GUARD-scaffold context. We searched for exact matches except for the three non-targeting GUARDs where we allowed a single mismatch as this would be without functional consequence. For reads with matching GUARD sequences, we aligned these to the oligo reference library and checked whether the correct corresponding context-off-target-context (79 bp) was aligned. For correctly assigned reads, we used a dpAlign from BioPerl for alignment and quantification of SNPs and indel frequency. Mutations occurring in the plasmid library were removed from the experimental data, as they are likely caused by NGS and cloning artefacts and not due to Cas9 or BE. All further processing and graphing of the data was performed in R and can be found here: (https://www.sanger.ac.uk/tool/crispr-guard-finder/ and https://github.com/MatthewACoelho/CRISPRGUARDFinder). In downstream analyses, we only considered GUARD RNA constructs with a minimum of 250 reads. Thus, coverage ranged between 250 and approximately 16,000 reads per GUARD RNA construct for ~600 unique constructs.

**CRISPR GUARD finder tool.** For each guide (Gon) we identified off-target sites with up to 5 mismatches using our own implementation of the method employed in the Sanger WGE website[35](https://www.sanger.ac.uk/htgt/wge/), enhanced with the calculation of the probability of the off-target[36]. We used version GRCh38 (hg38) of the human genome, and GRCm38 (mm10) of the mouse genome, and gene annotation obtained from Ensembl. For each off-target locus ($G_{off}$) with a probability above a threshold value, we scanned a user-defined proximal region of x bp for GUARD sequences ($G_{Guard}$) of a user-defined length (e.g., 14-nt or 15-nt) adjacent to an NRG PAM on either strand. The code is available from (https://www.sanger.ac.uk/tool/crispr-guard-finder/ and https://github.com/MatthewACoelho/CRISPRGUARDFinder).

**Reporting summary.** Further information on research design is available in the Nature Research Reporting Summary linked to this article.

## Data availability

All data generated or analysed during this study are included in this published article, its supplementary information files, and publicly available repositories. Sequencing data is available from the NCBI Sequence Read Archive database, accession SRP252950, BioProject accession PRJNA612602. Source data are provided with this paper. Any additional relevant data are available upon reasonable request. Source data are provided with this paper.

## Code availability

Code for the CRISPR GUARD Finder tool is freely available from: https://github.com/MatthewACoelho/CRISPRGUARDFinder. Code and analyses used to analyse sequencing data are available from: https://github.com/MatthewACoelho/CRISPRGUARDFinder.

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

## Acknowledgements

Thanks to Discovery Biology AstraZeneca for helpful discussion, Marcello Maresca for conceptual advice and Ian Barrett and Stephanie Ashenden for bioinformatic support. We thank Euan Gordon for Cas9 protein production. We thank Ohad Yogev and Ardi Liaunardy for critical reading of the manuscript. MC is a fellow of the AstraZeneca postdoc programme. This work was funded by AstraZeneca plc. We thank the Wellcome Sanger Institute for hosting the CRISPR GUARD Finder tool.

## Author contributions

M.A.C. and B.J.M.T. designed the study. M.A.C. carried out the experiments, analysed and interpreted the data. E.D.B. performed experiments for paper revisions. M.B. performed B.L.I. experiments. M.F. provided bioinformatics support for NGS data analysis and wrote the CRISPR GUARD Finder tool. M.A.C. wrote the CRISPR GUARD Finder Shiny app. S.L., M.E.C., M.M. and B.J.M.T. provided conceptual advice. M.A.C. and B.J.M.T. wrote the manuscript. All authors contributed to manuscript revision and review.

## Competing interests

The authors declare no competing interests.
