## [Peer Review File · Nature Communications]

Reviewer #1:

Remarks to the Author:

In this manuscript, the authors have designed a series of sgRNAs designed to reduce off-target gene editing by Cas9. The authors explore a series of RNA designs, characterizing the reduction of off-target editing rates both in test tubes as well as cell lines.

The paper had several strengths, which are listed below.

(1) The paper was written clearly.

(2) The figures, in general, were clear and easy to interpret.

(3) The authors did not overstate their results, or the implications thereof.

(4) I particularly enjoyed Sup Figure 3; whether reducing off-target 1 increased off-target 2-N was a very clever question to ask / answer.

There were also several aspects that should be improved before publication, which are also listed below.

(A) In Figure 2d and 2f, the experiment should be repeated with additional bioreps. It is difficult to interpret the data with the number of bioreps, and the resultant error bars.

(B) Please consider adding additional text to clarify what is plotted in Figure 2E. I was able to interpret it, but general readers with less experience in the field may have difficulty.

(C) This is a minor comment, and therefore isn't critical, but I would refrain from using the word 'damage' when describing editing events (line 131).

(D) I understand the need for pre-complexing the target sgRNA with active Cas9 and the GUARD RNA with the dCas9. It is, for this experimental setup, the appropriate combination. However, it would be good for the authors to discuss the inherent complexity of using this approach in the real world at the end of the paper.

(E) The authors correctly describe the potential risk for guide swapping. When the on-target event is mediated by any dCas9 variant (e.g., activation, base editing, etc.), even the shortened GUARD RNAs, if they bind to their own DNA, should still result in activation, base editing, etc., if the RNAs are swapped. If I am mistaken, please clarify more in the text. If I am correct, I would perform an additional experiment to address this issue.

Reviewer #2:

Remarks to the Author:

This paper describes a simple and useful method for suppressing off-target genome edits by delivering GUARD RNAs, which are truncated gRNAs having perfect complementarity to the potential off-target sites. Multiplexing was possible to block multiple off-target edits simultaneously, and the authors suggest some GUARD RNA design rules even though it should be elaborated further. Finally, the authors have shown that base editors work even with truncated gRNAs, which makes GUARD RNA less useful for suppressing off-target base editing. However, when carefully designed to avoid cytosine at the GUARD RNA binding sites, the method would be a valuable tool for base editing as well.

Overall, new findings and usefulness of the method make this paper suitable for publication in

Nature Communications. However, there are several issues that should be addressed.

1. One needs to identify potential off-target sites before starting the genome editing, and it is not trivial. As a solution, 'GUARDfinder' has been developed, but in many cases, there may be so many potential off-target sites, which in turn make it extremely difficult to screen all potential GUARD RNAs. The authors should describe these shortcomings and suggest potential solutions at the discussion part.

2. Some concepts are confusing, so they need to be described more clearly. In addition, there are many mistakes in the manuscript.

(Line 61) For the '15-nt + spacer' GUARD RNA, how did the authors design the 5 nt spacer sequence?

(Figure 2) It seems like that the same set of GUARD RNA was used for on-target and off-target cleavage in Figure 2c. In figure 2d, however, two types of GUARD RNAs was used (14 nt VEGFA GUARD and 14 nt CAVIN4 GUARD), and it is not clear what 'VEGFA GUARD' is. This should be explained clearly in the manuscript. Otherwise, the left panel of the figure 2d should show the data using CAVIN4 GUARD just as in figure 2c.

(Line 573) Information about statistics in figure 2f is not shown.

(Line 661) It is about figure 2e, not 2d.

In figure 2f and figure S2, it seem like that the same set of on-target and off-target sites were used, but the labels on the figures are different, which makes the manuscript confusing. In addition, it is not shown that which type of GUARD RNA was used in figure S2.

(Figure 3b) It would be better if the authors could show the directionality of gRNAs and GUARD RNAs more clearly (e.g. by indicating termini as 5' or 3'). Figure S6b seems clearer, so those kinds of supplementary figures could be added.

(Figure 4a) How was the % base editing calculated? Because there are several cytosines at the editing window, the way of calculation should be clearly described.

There are many mistakes in the manuscript when assigning the figure 5.

In figure 6b, the plots are showing % base editing, but in the figure legend, it is written as indel and SNP from BE. Clearer description is required.

In figure 6d, it should be described clearly what kind of measurements were compared for calculating P values.

In generating plots, the same axis labels should be used throughout the manuscript. For example, some plots have the label 'Base editing' while some have 'BE'.

3. In Figure S3, the authors investigated CAVIN4 as the off-target site for the plasmid-based genome editing experiments, but investigated the other off-targets sites for the RNP-based assay. Is there any reason for this? It would be better if the authors could show off-target effects in CAVIN4 to be able to compare the plasmid- and RNP-based delivery.

4. Sometimes, it is hard to read labels in the figure. Resolution and font size should be increased.

5. In table 1, are the data from GUARD RNA transfection showing the reads at the on-target (EMX1) amplicon or corresponding off-target regions?

6. In figure 4b, it seems like that HDLBPhiGC GUARD RNA did not decrease off-target base editing at the gRNA editing window while it did in the case of indel formation (Figure S4) at the same gRNA:GUARD RNA ratio, but gRNA-editable cytosines may not be open by the GUARD RNA considering the sequence context. Is there any explanation for this?

7. Figure 6f can be modified to give more information, not only to describe the factors affecting GUARD RNA performance, but also to suggest some design guidelines the authors have identified.

8. It would be good if the authors could show examples of using 'GUARDfinder' as a supplementary figure.

Reviewer #1 (Remarks to the Author):

In this manuscript, the authors have designed a series of sgRNAs designed to reduce off-target gene editing by Cas9. The authors explore a series of RNA designs, characterizing the reduction of off-target editing rates both in test tubes as well as cell lines.

The paper had several strengths, which are listed below.

- (1) The paper was written clearly.
- (2) The figures, in general, were clear and easy to interpret.
- (3) The authors did not overstate their results, or the implications thereof.
- (4) I particularly enjoyed Sup Figure 3; whether reducing off-target 1 increased off-target 2-N was a very clever question to ask / answer.

Thank you for your comments. It's good to hear that you found the paper clear and well written.

There were also several aspects that should be improved before publication, which are also listed below.

(A) In Figure 2d and 2f, the experiment should be repeated with additional bioreps. It is difficult to interpret the data with the number of bioreps, and the resultant error bars.

We have repeated Figure 2d with 2 additional biological replicates performed on separate days. This further increases the statistical significance of our findings and does not change any conclusions. Figure 2f also shows statistically significant differences, but unfortunately, due to the COVID pandemic, we are currently unable to re-perform additional NGS experiments at this time. We feel that additional repeats are unlikely to change the conclusions.

(B) Please consider adding additional text to clarify what is plotted in Figure 2E. I was able to interpret it, but general readers with less experience in the field may have difficulty.

Thank you. We have included additional text to describe Figure 2e (lines 94-106). In conjunction with the schematic of the experiment in Supplemental Figure 1, we hope this will be clearer for the reader.

(C) This is a minor comment, and therefore isn't critical, but I would refrain from using the word 'damage' when describing editing events (line 131).

Agreed. We have changed this in the text to "off-target editing".

(D) I understand the need for pre-complexing the target sgRNA with active Cas9 and the GUARD RNA with the dCas9. It is, for this experimental setup, the appropriate combination.

However, it would be good for the authors to discuss the inherent complexity of using this approach in the real world at the end of the paper.

Thank you for raising this point. We agree that this could be less practical in some contexts, but not prohibitive. We have included a short passage in the Discussion relating to this point (line 374-378).

(E) The authors correctly describe the potential risk for guide swapping. When the on-target event is mediated by any dCas9 variant (e.g., activation, base editing, etc.), even the shortened GUARD RNAs, if they bind to their own DNA, should still result in activation, base editing, etc., if the RNAs are swapped. If I am mistaken, please clarify more in the text. If I am correct, I would perform an additional experiment to address this issue.

Thank you for your comment. As you say, we discourage using 20-nt guides to block off-target editing by Cas9 due to guide swapping. However, in the context of base editing, if the GUARD RNA has cytosines within the base editing window there will be editing regardless of guide swapping (Figure 4b and 4c). Thus, we recommend avoiding cytosines in the protospacer of GUARD RNAs to avoid this (lines 339-348).

Reviewer #2 (Remarks to the Author):

This paper describes a simple and useful method for suppressing off-target genome edits by delivering GUARD RNAs, which are truncated gRNAs having perfect complementarity to the potential off-target sites. Multiplexing was possible to block multiple off-target edits simultaneously, and the authors suggest some GUARD RNA design rules even though it should be elaborated further. Finally, the authors have shown that base editors work even with truncated gRNAs, which makes GUARD RNA less useful for suppressing off-target base editing. However, when carefully designed to avoid cytosine at the GUARD RNA binding sites, the method would be a valuable tool for base editing as well.

Overall, new findings and usefulness of the method make this paper suitable for publication in Nature Communications. However, there are several issues that should be addressed.

1. One needs to identify potential off-target sites before starting the genome editing, and it is not trivial. As a solution, 'GUARDfinder' has been developed, but in many cases, there may be so many potential off-target sites, which in turn make it extremely difficult to screen all potential GUARD RNAs. The authors should describe these shortcomings and suggest potential solutions at the discussion part.

Thank you for your helpful comments and for raising this important point. We have included a passage in the Discussion in response (lines 360-370).

2. Some concepts are confusing, so they need to be described more clearly. In addition, there are many mistakes in the manuscript.

(Line 61) For the '15-nt + spacer' GUARD RNA, how did the authors design the 5 nt spacer sequence?

In this case, we chose the remaining five nucleotides such that there was no additional sequence complementarity to the off-target locus. Please see Supplementary Table 2 and lines 63 and 123-126.

(Figure 2) It seems like that the same set of GUARD RNA was used for on-target and off-target cleavage in Figure 2c. In figure 2d, however, two types of GUARD RNAs was used (14 nt VEGFA GUARD and 14 nt CAVIN4 GUARD), and it is not clear what 'VEGFA GUARD' is. This should be explained clearly in the manuscript. Otherwise, the left panel of the figure 2d should show the data using CAVIN4 GUARD just as in figure 2c.

Apologies for the confusion; we have clarified the text (line 90). In Figure 2d, we use a VEGFA GUARD RNA to test protection of the on-target site. It competes with the on-target VEGFA gRNA. The GUARD RNA is listed in Supplementary Table 2. CAVIN4 14-nt GUARD RNA does not have an impact on VEGFA on-target cutting (Figure 2d), and this is shown repeatedly in subsequent figures.

(Line 573) Information about statistics in figure 2f is not shown.

Sorry for this omission. We have corrected this.

(Line 661) It is about figure 2e, not 2d.

Thank you. We have corrected this.

In figure 2f and figure S2, it seem like that the same set of on-target and off-target sites were used, but the labels on the figures are different, which makes the manuscript confusing. In addition, it is not shown that which type of GUARD RNA was used in figure S2.

Sorry for this. We have made the labelling in Figure S2 consistent with the rest of the paper and made the GUARD RNA labelling explicit.

(Figure 3b) It would be better if the authors could show the directionality of gRNAs and GUARD RNAs more clearly (e.g. by indicating termini as 5' or 3'). Figure S6b seems clearer, so those kinds of supplementary figures could be added.

We agree, and have added the location of the GUARD RNA PAMs in Figure 3b to indicate directionality and make this consistent with the other figures. Please also see Supplementary Table 2 for the full sequence.

(Figure 4a) How was the % base editing calculated? Because there are several cytosines at the editing window, the way of calculation should be clearly described.

It is the cumulative editing frequency of the cytosines highlighted (those detected to be edited by NGS). We have now described this more clearly in the figure legend.

There are many mistakes in the manuscript when assigning the figure 5.

We have now corrected these typos.

In figure 6b, the plots are showing % base editing, but in the figure legend, it is written as indel and SNP from BE. Clearer description is required.

Thank you for your comment. The wording was confusing before and we have rectified this.

In figure 6d, it should be described clearly what kind of measurements were compared for calculating P values.

Thank you. We have described this more carefully in the revised figure legend for Figure 6d.

In generating plots, the same axis labels should be used throughout the manuscript. For example, some plots have the label 'Base editing' while some have 'BE'.

We have corrected this to make labelling consistent throughout.

3. In Figure S3, the authors investigated CAVIN4 as the off-target site for the plasmid-based

genome editing experiments, but investigated the other off-targets sites for the RNP-based assay. Is there any reason for this? It would be better if the authors could show off-target effects in CAVIN4 to be able to compare the plasmid- and RNP-based delivery.

The majority of experiments were performed using Cas9-expressing cell lines rather than plasmid-based delivery. We have performed a series of experiments comparing RNP delivery (Figure S3) with and Cas9-expressing cells transfected with synthetic gRNA (Figure S5). Both experiments included VEGFA on-target and GUARDS protecting TENT4A. However, there is a challenge in comparing the results due to the vastly differing on-target efficiencies achieved, with >50% with Cas9 expressing cells and a modest 12% with RNP delivery. Both delivery systems showed a broadly similar 2-3 fold reduction in TENT4A editing when GUARDS were included. It should be noted that both sets of experiments were conducted with equimolar concentrations of gRNA:GUARDS and so increasing GUARD concentration will likely improve the protection achieved.

4. Sometimes, it is hard to read labels in the figure. Resolution and font size should be increased.

Apologies. We have made every effort to improve readability and figure resolution in the revised version.

5. In table 1, are the data from GUARD RNA transfection showing the reads at the on-target (EMX1) amplicon or corresponding off-target regions?

They show the regions targeted by the given GUARD RNA. We have now indicated this in the table to improve clarity.

6. In figure 4b, it seems like that HDLBPhiGC GUARD RNA did not decrease off-target base editing at the gRNA editing window while it did in the case of indel formation (Figure S4) at the same gRNA:GUARD RNA ratio, but gRNA-editable cytosines may not be open by the GUARD RNA considering the sequence context. Is there any explanation for this?

This is a good point, and we do not have an absolute explanation for this, but can speculate. We think it may be related to the expression levels of BE3 vs Cas9 in the cells, with lower Cas9 protein levels possibly creating a context more amenable to CRISPR GUARD (i.e. off-target editing is lower to begin with).

7. Figure 6f can be modified to give more information, not only to describe the factors affecting GUARD RNA performance, but also to suggest some design guidelines the authors have identified.

We have made these changes and agree that it makes the summary presented in Figure 6f more useful for the reader.

8. It would be good if the authors could show examples of using 'GUARDfinder' as a supplementary figure.

Thank you for this point. We have considerably improved the CRISPR GUARD Finder tool since our first submission, and provide an online server for this, which allows ease of use: <https://www.sanger.ac.uk/science/tools/crisprguardfinder/crisprguardfinder/>
We have included an example of using the GUARD Finder tool in the revised Supplement. Please see Supplementary Fig. 12 for an example.

Reviewers' Comments:

Reviewer #1:

Remarks to the Author:

The authors have addressed all my concerns. I find the paper suitable for publication.

Congratulations on the interesting work!

Reviewer #2:

Remarks to the Author:

Overall, the authors revised and improved the manuscript significantly. The following minor points may improve the manuscript further.

How did the authors design NT GUARDs for all experiments? Their sequences should be described in the supplementary tables.

In all plots, authors should not use statistical tests when 'n' is less than three.

(Line 167) This reviewer does not think that the new high-GC GUARD is superior in this example. Previous and new GUARDs gave very small differences (Supplementary figure 4), and the new GUARD has a longer overlap with the seed region. Thus, it is hard to say that GC content is the only reason for different results. Based on supplementary figure 11, GC content should affect the GUARD function, but it may not be the case here. The manuscript needs to be revised accordingly.

(Supplementary figure 6b) The legend is referring figure 1a, but probably it is figure 1. Even though readers may be able to understand the directionality of the sequence, the convention is to match (left to right) to (5' to 3'). In the figure, however, some are with 5' to 3' and others with 3' to 5'. The directionality should be indicated clearly.

(Figure 4b and c) When comparing two plots for MFAP1 GUARD, the rate of off-target editing coming from GUARD is much higher when GUARD is delivered with on-target gRNA. Is there any explanation for this?

(Figure 6f) 'High GC content of > 50%' may not be included here or modified because data at supplementary figure 11a does not support the cut-off of 50%. From the data, GC content higher than ~35% seems more appropriate.

(Line 321) Supplementary table 4 is not present.

(Figure 2) It is written in the legend that figure 2a is showing mean \pm SD from two experiments, and 2b from three experiments. However, 2a seems to show the result from a single experiment, and 2b is a schematic. These errors should be corrected.

REVIEWERS' COMMENTS:

Reviewer #1 (Remarks to the Author):

The authors have addressed all my concerns. I find the paper suitable for publication.

Congratulations on the interesting work!

Reviewer #2 (Remarks to the Author):

Overall, the authors revised and improved the manuscript significantly. The following minor points may improve the manuscript further.

How did the authors design NT GUARDs for all experiments? Their sequences should be described in the supplementary tables.

We have clarified this point (line 487), and add all the NT controls sequences in Supplementary Table 2. We took published NT guide RNA sequences (Doench *et al. Nat. Biotechnol.*, 2016) and used truncated versions, verifying with BLAST that they had minimal complementarity to the human genome.

In all plots, authors should not use statistical tests when 'n' is less than three.

We have changed this accordingly.

(Line 167) This reviewer does not think that the new high-GC GUARD is superior in this example. Previous and new GUARDs gave very small differences (Supplementary figure 4), and the new GUARD has a longer overlap with the seed region. Thus, it is hard to say that GC content is the only reason for different results. Based on supplementary figure 11, GC content should affect the GUARD function, but it may not be the case here. The manuscript needs to be revised accordingly.

We understand this reservation. There is a modest decrease in off-target editing with the redesigned GUARD RNA, and as the reviewer points out, data from Supplementary Figure 11 adds supporting evidence for GC-content being important. Please note that we do not conclude from Supplementary Figure 4 alone that higher GC-content will improve GUARD RNA performance, nor do we feel we overstate our conclusions. This being said, we have carefully reviewed and changed the text relating to Supplementary Figure 4 accordingly (line 167, and below, underlined). We have also modified the figure legend appropriately.

"Introduction of a revised GUARD RNA with higher GC-content, length and increased overlap of the gRNA seed region, achieved only modestly improved protection (Supplementary Fig. 4). We only observed protection at a 5:1 molar ratio of GUARD RNA to gRNA, and no protection at a 1:1 ratio. Taken together, these data suggest that some off-target loci are more amenable to protection by CRISPR GUARD than others, indicating more systematic investigation is warranted. This is likely contingent on the relative affinities of the GUARD RNA and the cognate gRNA for the off-target region."

(Supplementary figure 6b) The legend is referring figure 1a, but probably it is figure 1.

Corrected this to "Figure 1".

Even though readers may be able to understand the directionality of the sequence, the convention is to match (left to right) to (5' to 3'). In the figure, however, some are with 5' to 3' and others with 3' to 5'. The directionality should be indicated clearly.

Thank you for this comment. We feel that our representation of the GUARD RNAs relative to the gRNA sequence is useful. We have now added 5' and 3' annotation to improve clarity.

(Figure 4b and c) When comparing two plots for MFAP1 GUARD, the rate of off-target editing coming from GUARD is much higher when GUARD is delivered with on-target gRNA. Is there any explanation for this?

Yes – we describe this on line 227 (please see below). In essence, on-target guide causes some editing and nicks the DNA, so when combined with the editing GUARD RNA, more edits are retained after DNA repair than either in isolation.

"It is likely that the low level of base editing observed with short GUARD RNAs alone is due to the lack of nickase activity, which is analogous to editing with base editor 2 versions²⁸."

(Figure 6f) 'High GC content of > 50%' may not be included here or modified because data at supplementary figure 11a does not support the cut-off of 50%. From the data, GC content higher than ~35% seems more appropriate.

We have changed the schematic in Figure 6f to say "High GC content", and removed ">50%".

(Line 321) Supplementary table 4 is not present.

My apologies. We have removed reference to Table 4, which does not exist. All information including the CRISPR GUARD Finder output is included in Supplementary Figure 12.

(Figure 2) It is written in the legend that figure 2a is showing mean \pm SD from two experiments, and 2b from three experiments. However, 2a seems to show the result from a single experiment, and 2b is a schematic. These errors should be corrected.

We have corrected the legend. Thank you for pointing out this error. For Figure 2a, data are representative of two independent experiments. Source data are provided as a Source Data File.